# Molecular Interactions of the Long Noncoding RNA NEAT1 in Cancer

**DOI:** 10.3390/cancers14164009

**Published:** 2022-08-19

**Authors:** Jingtao Gu, Bo Zhang, Rui An, Weikun Qian, Liang Han, Wanxing Duan, Zheng Wang, Qingyong Ma

**Affiliations:** Department of Hepatobiliary Surgery, The First Affiliated Hospital of Xi’an Jiaotong University, Xi’an 710061, China

**Keywords:** NEAT1, cancer, paraspeckle, transcription, microRNA

## Abstract

**Simple Summary:**

In this review, we present current knowledge of LncRNA NEAT1 in the development of malignant diseases. First, we illustrate the structure and functions of NEAT1 in cancer. Then we explore the interaction between NEAT1 and other molecules and analyze the impact of this interaction on cancer progression. Finally, we summarize the regulation of NEAT1 and the function of NEAT1-containing exosomes. Elucidating the molecular interaction of NEAT1 may shed light on future treatment of cancer.

**Abstract:**

As one of the best-studied long noncoding RNAs, nuclear paraspeckle assembly transcript 1 (NEAT1) plays a pivotal role in the progression of cancers. NEAT1, especially its isoform NEAT1-1, facilitates the growth and metastasis of various cancers, excluding acute promyelocytic leukemia. NEAT1 can be elevated via transcriptional activation or stability alteration in cancers changing the aggressive phenotype of cancer cells. NEAT1 can also be secreted from other cells and be delivered to cancer cells through exosomes. Hence, elucidating the molecular interaction of NEAT1 may shed light on the future treatment of cancer. Herein, we review the molecular function of NEAT1 in cancer progression, and explain how NEAT1 interacts with RNAs, proteins, and DNA promoter regions to upregulate tumorigenic factors.

## 1. Introduction

Approximately 80% of the human genome is transcribed into RNA, but less than 2% can be translated into proteins [1]. Therefore, the majority of transcribed RNA belongs to noncoding RNAs (ncRNAs), of which those with lengths over 200 nucleotides are termed long noncoding RNAs (lncRNAs) [2].

According to various transcription patterns, lncRNAs can be categorized as enhancer lncRNAs, promoter upstream transcripts (PROMPTs), exon or intron sense-overlapping lncRNAs, long intergenic ncRNAs (lincRNAs), bidirectional lncRNAs, and natural antisense transcripts (NATs) (Figure 1A) [2,3]. Among the various types of lncRNAs, the best-studied are the lincRNAs transcribed from DNA sequences between two genes [3].

Nuclear paraspeckle assembly transcript 1 (NEAT1), which is transcribed from the multiple endocrine neoplasia (*MEN*) locus on chromosome 11q13.1, is a well-studied lincRNA in the field of cancer diseases [4,5]. Since NEAT1 belongs to the family of nucleotides, it has the ability to interact with other nucleotides, including mRNA, miRNA, or DNA, to change the expression of downstream proteins. In addition, the secondary structure of NEAT1 can capture some proteins and change their stability or molecular function [6].

Studies have shown that NEAT1 contributes to the progression of many cancers, including breast cancer, lung cancer, hepatocellular carcinoma, ovarian cancer, and prostate cancer [7,8,9,10]. A higher level of NEAT1 almost always indicates a poorer survival rate in cancer patients [11,12,13]. NEAT1 can also reduce cancers’ sensitivity to chemotherapy or radiotherapy [14,15]. However, NEAT1 overexpression seems to be a protective factor in acute myeloid leukemia (AML), and lower NEAT1 levels are responsible for differentiation disorders in AML [16,17].

Previous reviews of NEAT1 mainly depicted the interaction between NEAT1 and miRNAs and discussed other mechanisms in very short paragraphs. In our review, however, we describe the less-concerned DNA binding capacity of NEAT1 and place more emphasis on RNA-binding proteins. First, we illustrate the structure and functions of NEAT1 in cancer. Then, we explore the interaction between NEAT1 and other molecules and analyze the impact of this interaction on cancer progression. Finally, we summarize the regulation of NEAT1 and the function of NEAT1-containing exosomes. As an important diagnostic and therapeutic marker, further elucidation of the mechanisms of NEAT1 in cancer progression will undoubtedly benefit clinical work [18].

## 2. Structure of NEAT1 and Paraspeckles

NEAT1 has two transcription variants, namely, NEAT1-1 (3756 nt) and NEAT1-2 (22,743 nt) [19] (Figure 1B). Specifically, the shorter transcript NEAT1-1 (also known as MENepsilon) is stabilized by a canonical poly (A) tail at the 3′ end, while the longer transcript NEAT1-2 (also known as MENbeta) does not have a poly (A) tail and is stabilized by a triple helical structure at the 3′ end [20,21]. These two isoforms are produced by alternative 3′-end processing, and the sequence of NEAT1-1 can be regarded as part of NEAT1-2 [19]. Accordingly, siRNA, shRNA, and PCR primers designed to knockdown or detect NEAT1-1 actually target both of the isoforms of NEAT1, namely, NEAT1-1 and NEAT1-2. When the polyadenylation of NEAT1-1 is inhibited, the transcription process continues at the gene locus, forming the longer transcript NEAT1-2 [19,22]. Interestingly, the pre-miR-612 gene is also located at the MEN locus and shares an overlapping sequence with NEAT1-2. Mature miR-612 is a tumor-suppressive gene generated by alternative splicing, and the balance between the two NEAT1 transcripts and miR-612 can be regulated by polypyrimidine tract binding protein 3 (PTBP3) [23]. The two NEAT1 isoforms and certain RNA-binding proteins (RBPs) together form a pivotal nuclear structure named a paraspeckle (Figure 1C). NEAT1-2 plays a more crucial role in the formation of paraspeckles, while NEAT1-1 has been predicted to bind with the 3′ end of NEAT1-2 due to their complementary base pairs [24]. More than 50 RBPs, including paraspeckle component 1 (PSPC1), non-POU domain-containing octamer-binding protein (NONO/p54nrb), and splicing factor proline/glutamine-rich (SFPQ/PSF), can be members of paraspeckles [25,26].

Paraspeckles exert their functions mainly through the retention or sequestration of RNAs or proteins [27]. Paraspeckles can capture A-to-I–edited mRNAs, limiting their export from the nucleus [28]. Zhang et al. showed that multiple I residues in the edited RNA can form a complex with PSF, NONO, and matrin 3, resulting in the retention of an RNP complex in the nucleus [29]. Generally, dsRNA-dependent adenosine deaminases (ADARs) can convert adenosine (A) into inosine (I), which will be identified as guanosine (G) in the translation process, forming a different peptide if this is not a synonymous substitution [30]. Even though there is a synonymous substitution, which does not change the type of coding proteins, it can affect the degradation velocity of mRNAs, subsequently regulating the expression level of the corresponding proteins [31]. Hence, paraspeckles can prevent the translation of these A-to-I–edited mRNAs to a certain extent. In addition, the formation or unraveling of paraspeckles will change the amount of paraspeckle component proteins (PSPs), such as PSPC1, SFPQ, and NONO, sequestered in the paraspeckles [25]. For instance, a lower number of paraspeckles sequester less SFPQ; thus, the released SFPQ will relocate to the promoter region of the IL-8 gene locus, inhibiting the transcription of IL-8 mRNA in HeLa cells [32]. Paraspeckles can also promote the immune escape of hepatocellular carcinoma by sequestrating IFNGR-1 mRNA, inhibiting IFN-γ induced T-cell killing effects [33].

## 3. NEAT1 Sponges microRNAs and Stabilizes mRNAs

NEAT1 can bind to various microRNAs (miRNAs/miRs) and sponged miRNAs cannot combine with mRNA in the RNA-induced silencing complex (RISC) [34]. If the 3′ untranslated region (3′UTR) of mRNA and miRNA are perfectly complementary, the resulting heterodimers can be degraded in the RISC [12]. In contrast, the sponged dysfunctional miRNA prevents the degradation of mRNA, causing larger amounts of mRNA to be translated into functional proteins [5]. This is the most common mechanism identified by researchers who have attempted to elucidate the functions of NEAT1 in cancer diseases.

As an endogenous competing RNA (ceRNA), NEAT1 can facilitate the progression of many cancers (Table 1). For instance, NEAT1 can sponge miR-302a-3p, which is a tumor-suppressive molecule that suppresses the translation of RELA mRNA, thereby upregulating RELA and increasing the migration and proliferation capacity of pancreatic cancer cells [12]. Notably, NEAT1 has two transcripts, and some miRNAs can only interact with NEAT1-2, such as miR-491 and miR-106b-5p, which promote the metastasis and growth of thyroid cancer [35]. In this section, we mainly discuss the ceRNA network in various cancers.

Breast cancer is the leading cause of cancer death in the female population all over the world and its incidence rates are gradually increasing [36]. The expression of NEAT1 is elevated in breast cancer cell lines, including MCF-7, MDA-MB-453, MDA-MB-231, and SKBR3 cells, compared with normal mammary epithelial cells such as MCF-10A [37]. Moreover, the expression of NEAT1 is higher in cancer tissues compared with the normal adjacent tissues [38]. NEAT1 can upregulate many oncogenic factors, such as KLF12, Cyclin D1, ZEB1, and HMGA2, which subsequently promote the proliferation and metastasis of breast cancer [37,38,39,40,41,42].

Colorectal cancer is an important cause of global cancer-related death due to its late diagnosis and rapid metastasis [13,43]. One of the biggest obstacles to treating colorectal cancer is the lack of a molecular biomarker. Studies have shown that NEAT1 expression is upregulated in colorectal cancer tissues, and a higher level of NEAT1 contributes to the poor overall survival and disease-free survival [44,45]. Moreover, NEAT1 can also promote the proliferation and metastasis of colorectal cancer in vivo and in vitro [44,45]. NEAT1 can elevate the expression of some growth factors and transcription factors such as IGF2, GDNF, and BACH1, which facilitate the progression of colorectal cancer [44,46,47].

Gastric cancer is the third leading cause of cancer-related death worldwide [48]. Although early treatment of gastric cancer has achieved significant success, early diagnosis is still difficult; most patients present with an advanced stage at the time of diagnosis, causing a poor survival rate [49,50,51]. Hence, identifying novel biomarkers is important to improve the diagnosis and treatment of gastric cancer. NEAT1 is significantly upregulated in human gastric cancer cells, including BGC823, SGC-7901, AGS, MGC803, and MKN28 cells, compared with normal gastric epithelial cells such as GES-1 [52]. Studies have shown that NEAT1 can upregulate the expression of ABCC4, JAG1, SOX11, COX-2, etc. These factors will finally promote the proliferation, migration, and invasion of gastric cancer [49,53,54,55].

Glioma is the most prevalent primary central nervous system tumor, which has a high postoperative recurrence and mortality rate [56]. Although glioma can be treated by surgery, it is always hard to excise completely [57]. Studies have shown that NEAT1 is upregulated in glioma tissues compared with adjacent normal tissues [57,58]. NEAT1 can elevate the expression of many molecules, including DNMT1, CCT6A, CDK6, SOX2, and c-MET in glioma cells. Therefore, a higher level of NEAT1 can promote the proliferation, migration, and epithelial–mesenchymal transition (EMT) process in glioma cells [56,57,58,59,60].

Hepatocellular carcinoma causes approximately 600,000 deaths each year. It is the sixth most common cancer all over the world and the third leading cause of cancer-related death [61,62]. Chronic hepatitis B or C virus infections are the pivotal development factors of hepatocellular carcinoma [63]. Frequent intrahepatic and extrahepatic metastases contribute to the low resectability and high recurrence rate even in the early stage of this disease [64]. NEAT1 is upregulated in hepatocellular carcinoma tissues and cell lines compared with normal adjacent tissues and normal cell lines [8]. NEAT1 can elevate the expression of many factors, including SMO, LAGE3, AKT2, TGF-β1, and STAT3, which can increase the proliferation and metastasis both in vivo and in vitro, accelerating the progression of hepatocellular carcinoma [61,63,64,65,66]. NEAT1 can also sponge miR-124-3p, elevating the expression of ATGL1, which promotes the lipolysis and cancer progression in hepatocellular carcinoma [67].

Lung cancer is one of the most prevalent malignant tumors and the leading cause of cancer-related death worldwide [68]. About 80% of lung cancers are non-small cell lung cancers (NSCLC) [69]. Moreover, the 5-year survival rate of NSCLC patients still remains unsatisfactory. Studies have shown that the 5-year survival rate of late NSCLC patients is less than 15% [70]. The upregulation of NEAT1 in lung cancer tissues has been observed in several studies [68,69,71]. NEAT1 can promote the proliferation and metastasis of NSCLC in vivo and in vitro by upregulating the expression of many oncogenes, including NAUK1, SULF1, SOX9, and E2F3 [68,69,70,71]. Higher NEAT1 levels are also associated with more lymph node metastasis, higher TNM grades, and a poor overall survival in patients [70].

Ovarian cancer is a common malignant tumor of the female reproductive system and has a high mortality rate [10]. As the early symptoms are not obvious, the disease onset is hidden, and metastasis occurs earlier. About 47% of women diagnosed with ovarian cancer can survive 5 years after diagnosis [72]. Yin et al. [73] reported that NEAT1 is upregulated in ovarian tumors compared with the adjacent tissues. Moreover, the expression level of NEAT1 is higher in ovarian cancer cell lines including ES2, A2780, HO8910, and SKOV3 than the normal ovarian epithelial cell line IOSE80. Clinical data showed that highly-expressed NEAT1 is closely correlated with a shorter survival rate, a poor differentiation degree, larger tumors, an advanced FIGO stage, and significant peritoneal metastasis in patients with ovarian cancer [10]. NEAT1 can upregulate the expression of TJP3, MEST, and ROCK1 in ovarian cancer cell lines, subsequently promoting the progression of ovarian cancer [10,72,73]. NEAT1 can also regulate glucose metabolism to promote ovarian cancer’s progression. For instance, the knockdown of NEAT1 has been shown to upregulate miR-4500 and downregulate BZW1, subsequently inhibiting the glycolysis of ovarian cancer [74].

Prostate cancer is the second most common malignant tumor and the fifth leading cause of cancer-related death among the male population worldwide [75]. Guo et al. [75] found that an overexpression of NEAT1 is correlated with an advanced clinical stage of disease. After detecting the expression of NEAT1 in tissues and cell lines, they observed that NEAT1 was significantly upregulated in prostate cancer tissues and cell lines compared with normal tissues and the normal prostate epithelial cell line. A study showed that NEAT1 can promote the proliferation and invasion of ovarian cancer cells via the upregulation of HMGA2 [75]. Moreover, NEAT1 can elevate the expression ACSL4, enhancing docetaxel resistance in PCa cells in vitro and in vivo [76]. Hence, NEAT1 promotes the progression of prostate cancer and may be a therapeutic biomarker in clinical studies.

The contribution of NEAT1 to cancer progression has also been studied in other cancers, including cervical cancer [77], cholangiocarcinoma [78], endometrial cancer [79], esophageal carcinoma [80], gallbladder cancer [81], glioblastoma [82], Hodgkin’s lymphoma [83], laryngeal carcinoma [84], melanoma [85], multiple myeloma [11], nasopharyngeal carcinoma [86], osteosarcoma [87], pancreatic carcinoma [88], retinoblastoma [89], thyroid cancer [90], and tongue carcinoma [91]. In those cancers, NEAT1 plays a pro-proliferative role and promotes the progression of cancer by upregulating the expression of some oncogenes or downregulating the expression of some tumor suppressor genes. For example, a study showed that the knockdown of NEAT1 can sensitize 5-FU resistance in cervical cancer and inhibit its glycolysis via the miR-34a/LDHA axis [92].

Interestingly, NEAT1 exerts an opposite function in acute myeloid leukemia. As an important hematologic malignant disease, acute myeloid leukemia is marked by the abnormal abundance of clonal myeloid progenitor cells in the bone marrow and the inhibition of normal hematopoiesis [16]. NEAT1 upregulation suppresses cell growth, migration, and invasion but enhances the apoptosis of acute myeloid leukemia cells [17]. Furthermore, the expression of NEAT1 is downregulated in acute myeloid tissues compared with normal tissues [93]. In patients with acute myeloid leukemia, NEAT1 is downregulated and subsequently decreases the expression of CREBRF, promoting the progression of the disease [17].

Interestingly, NEAT1 can directly interact with mRNA to affect its stability without the participation of miRNA. NEAT1 can bind and stabilize TNFRSF1B mRNA, contributing to the NF-κB signaling pathway [127]. Feng et al. [128] suggested that NEAT1 can also increase the stability of ELF3 mRNA by increasing its N6-methyladenosine (m6A) modification, promoting the proliferation and metastasis of pancreatic cancer.

## 4. NEAT1 Recruits Transcription Factors, Enzymes and Other Proteins

Due to the relatively few existing studies of NEAT1-binding proteins involved in cancers, we include several RBPs involved in other diseases to fully illustrate this mechanism. When LncRNA NEAT1 and some specific proteins together form a complex, the functions of NEAT1 can be divided into three types: a guide, decoy, and scaffold [18]. In this review, NEAT1-binding proteins are categorized as enzymes, transcription factors, and receptors (Figure 2).

## 5. NEAT1-Binding Enzymes

Many RBPs can bind to NEAT1 to regulate gene expression, and they are categorized in Table 2. First, we will discuss the NEAT1-binding enzymes involved in cancer progression. NEAT1-binding enzymes mainly include DNA methyltransferase, histone methylase, and E3 ubiquitin protein ligase. Hence, epigenetic mechanisms play a pivotal role in the regulation of gene expression. For instance, NEAT1 can bind to DNA methyltransferase 1 (DNMT1) to facilitate the growth and metastasis of lung cancer by increasing DNA methylation in the promoter region of *CGAS*, *STING,* and *P53* [129]. In addition, the knockdown of NEAT1 also promoted T cell infiltration in a xenograft tumor model [129]. EZH2, which is a subunit of the histone methylation transferase polycomb repressive complex 2 (PRC2), can also interact with NEAT1 [130]. EZH2 has the capacity to promote histone 3 lysine 27 trimethylation (H3K27me3), which is a transcriptional repressor of many tumor suppressor genes, such as GSK3B, ICAT, and Axin2 [131]. When NEAT1 recruits EZH2 to the promoter region of the genome locus, the repression of these genes contributes to the progression of cancers such as glioblastoma [131]. NEAT1 can also relocate EZH2 to the promoter regions of many other genes, such as *P21*, *LATS2*, *MYOG*, *MYH4,* and *TNNI2* [130,132].

NEAT1 can also sequester WDR5, which is a component of the mixed-lineage leukemia (MLL) H3K4 methylase complex responsible for histone 3 lysine 4 trimethylation (H3K4me3) within the promoter regions of SM-specific genes [133]. Since H3K4me3 is an active histone modification, the upregulation of NEAT1 can inhibit the expression of SM-specific genes—such as SM22a and calponin—in smooth muscle cells via the sequestration of WDR5 [133]. Moreover, the authors found that a depletion of NEAT1 can increase histone 3 lysine 9 acetylation (H3K9ac) and decrease H3K27me3 modification within the SM22a and calponin promoter regions, promoting gene transcription. Another study showed that NEAT1 can interact with the P300/CBP complex to affect the histone modification of H3K27 [134]. The inhibition of NEAT1 decreased H3K27ac but increased histone 3 lysine 27 crotonylation (H3k27Cro), downregulating the expression of endocytosis-related genes and aggravating Alzheimer’s disease [134]. Similarly, BRG-1, which is a subunit of SWI/SNF, can interact with NEAT1; thus, BRG-1 can be recruited to the promoter region of *GADD45A*, increasing the level of H3K27me3 and decreasing the level of H3K4me3 [135]. As a result, the transcription of GADD45A is inactivated, and gastric cancer is aggravated [135].

TRAF6, which is an E3 ubiquitin protein ligase, can also be pulled down by NEAT1 [139]. The autoubiquitination of TRAF6 can be inhibited by NEAT1, and an elevated TRAF6 level alleviates acute-on-chronic liver failure [139]. Moreover, NEAT1 can interact with another E3 ubiquitin protein ligase, NEDD4L, promoting the ubiquitination-mediated degradation of PINK1, which plays a crucial role in the maintenance of mitochondrial integrity via mitophagy [138]. As a consequence, downregulated PINK1 impairs mitochondrial function and promotes Alzheimer’s disease [138]. NEAT1 can also stabilize STAT3 by inhibiting the ubiquitination-mediated degradation of STAT3 in CD4+ T cells [140].

Recently, NEAT1 has also been shown to interact with cyclin-dependent kinase and glucose metabolism-related enzymes. In prostate cancer, NEAT1 can bind with cyclinL1/CDK19 to promote cancer metastasis [137]. In breast cancer, NEAT1 can bind and form a scaffold bridge for the assembly of phosphoglycerate kinase (PGK), phosphoglycerate mutase (PGAM), and enolase (ENO), elevating glycolysis and cancer metastasis [136].

## 6. NEAT1-Binding Transcription Factors

NEAT1 is associated with many transcription factors and relocates them to or sequesters them from the promoter regions of specific genes, activating or inactivating gene transcription [137,142]. For example, NEAT1 can interact with CDC5L and recruit it to the promoter region of *AGRN*, facilitating prostate cancer’s progression [141]. Furthermore, NEAT1 can exert a bridge function between two proteins, and the junction between two proteins can be inhibited by RNase. For example, NEAT1 can form an indispensable bridge between SIN3A and FOXN3. FOXN3 only has a transcription-activating domain and cannot bind to the DNA promoter region, whereas SIN3A only has a DNA-binding domain [36]. Therefore, the combination of these two proteins is essential for normal transcription. The SIN3A–NEAT1–FOXN3 complex ultimately regulates the downstream transcription of GATA3 and TJP1, enhancing epithelial-to-mesenchymal transition (EMT) in breast cancer [36]. Nuclear receptors are considered a special type of transcription factor that translocate from the cellular membrane to the nucleus after pairing with ligands. NEAT1 can interact with the nuclear receptor ERa, and then the recruited ERa binds to the promoter region of *AQP7*, promoting steatosis in hepatic cancer cells [143].

Another important factor is SFPQ, which can bind to the promoter region of *IL-8* and *ADARB2* to downregulate the transcription of IL-8 or upregulate the transcription of ADARB2 [20,32]. SFPQ is a classical paraspeckle protein with two RNA recognition motifs (RRMs), and NEAT1 can sequester it from the promoter regions of genes, inhibiting its transcriptional function [32]. Lee et al. suggested that there is a putative DNA-binding domain in the structure of SFPQ, but the current studies are not sufficient to support this molecule as a transcription factor [149]. SFPQ can also promote miRNA processing by binding with Drosha/DGCR8 in the NEAT1 scaffold [144]. Strikingly, SFPQ can regulate mRNA translation. NEAT1 can sequester SFPQ and block its binding with the internal ribosome entry segment (IRES) of c-Myc mRNA, thus inhibiting mRNA translation [145]. Interestingly, the paraspeckle has more than 50 types of proteins, including SFPQ, NONO, and FUS/TLS, which means that these proteins can interact with NEAT1 in the nucleus. However, how this interaction contributes to cancer progression still needs further study [150]. Additionally, transcription coactivators can also interact with NEAT1. For instance, DDX5 can promote colorectal cancer metastasis by interacting with NEAT1 and β-catenin [13].

Finally, certain proteins related to the spliceosome, such as SRSF5 and U2AF65, can interact with NEAT1. After binding with NEAT1, SRSF5 can regulate PPAR-γ mRNA splicing, increasing the PPAR-γ2 isoform expression in adipocytes [147]. U2AF65-mediated hnRNPA2 elevation also depends on the existence of NEAT1, facilitating the progression of hepatocellular carcinoma [148].

## 7. NEAT1 Interacts with DNA Sequences to Regulate Transcription

When we indicated that NEAT1 can recruit proteins such as EZH2 to the promoter regions of specific genes, we assumed that NEAT1 had a guide function. However, how NEAT1 guides EZH2 to the promoter region of specific genes has not been discussed. It is difficult to determine whether NEAT1 exerts a guidance function or only provides a scaffold and translocates with EZH2.

With designed biotin-labeled primer sets for NEAT1, the NEAT1-binding DNA sequence can be pulled down by streptavidin magnetic beads using many methods, such as chromatin isolation by RNA purification (CHIRP), the capture hybridization analysis of RNA targets (CHART), and RNA affinity purification (RAP) [151].

Researchers have suggested many NEAT1-binding elements after the sequencing DNA segments precipitated in this way. CHART-seq suggested that NEAT1 can bind to the transcriptional start sites and transcriptional termination sites of its target genes [151]. Moreover, Wen et al. predicted a binding site between NEAT1 and the promoter region of *RUNX2* through an algorithm analyzing the triplex binding capacity of NEAT1 to DNA duplexes. Then, the predicted binding site was verified by CHIRP-qPCR using the predicted primers. CDK19 and cyclinL1 are responsible for the phosphorylation of RNA pol II to promote the transcription of genes [137]. However, this study showed that the junction of the *RUNX2* promoter region with CDK19 or Cyclin L1 requires the existence of NEAT1, supporting the notion of a direct binding between NEAT1 and DNA sequences [137].

Similar to NEAT1, lncRNA HOTAIR can also recruit EZH2 to the promoter regions of many genes to regulate gene transcription. Its recruitment capacity was further explored by CHIRP-seq, and the analysis of HOTAIR binding sequences revealed an enrichment of GA-rich motifs [152]. After the EZH2 depletion, however, the profile of HOTAIR occupancy on the genome locus was not altered [152]. Therefore, HOTAIR can bind to DNA sequences without EZH2 and can directly bind to GA-rich motifs in DNA segments.

RNA–DNA interactions are rarely studied in relation to NEAT1, and future studies are still needed to explore the corresponding detailed mechanism.

## 8. Factors That Regulate NEAT1 Expression

The expression of NEAT1 is mainly regulated in three ways: transcriptional regulation, alternative splicing, and stability alteration. Studies have shown that some transcription factors can directly bind to the promoter region of NEAT1 to regulate its expression. Specifically, NF-κB, OCT4, HIF-2a, and c-Myc can promote the transcription of *NEAT1* in pancreatic cancer, lung cancer, breast cancer, and colorectal cancer, respectively [12,45,153,154]. Subsequently, the elevation of NEAT1 contributes to cancer progression (Figure 3A). Moreover, STAT3 can upregulate the expression of NEAT1 in HeLa cells to increase the viral replication of herpes simplex virus-1 (HSV-1) [142]. In contrast, some transcription factors can inhibit the expression of NEAT1 in cancer cells and immune cells. Lo et al. showed that BRCA1 can downregulate the expression of NEAT1 to suppress the progression of breast cancer [96]. NEAT1 can also be inhibited by E2F1, which decreases the H3K27ac level at the *NEAT1* promoter, finally inducing a tolerogenic phenotype of dendritic cells [155].

In addition, some molecules can regulate the ratio of the two isoforms of NEAT1, namely, the long transcript NEAT1-2 and the short transcript NEAT1-1. When the polyadenylation of NEAT1-1 is activated, the transcription site cannot cross the end of NEAT1-1 (Figure 3B). Consequently, the expression level of NEAT1-1 will be elevated, and the generation of NEAT1-2 will be inhibited. Specifically, TDP-43 can increase polyadenylation levels, upregulating NEAT1-1, and downregulating NEAT1-2 in mouse embryonic cells. Finally, the formation of paraspeckles is repressed and pluripotent cells can maintain their pluripotency and embryonic patterning [156]. In mouse embryonic fibroblasts, CPSF and NUDT21 can form a heterodimer to promote alternative polyadenylation (APA); the formed poly-A tail can stabilize NEAT1-1 and inhibit the elongation of NEAT1-2 [150]. However, HNRNPK and ARS2 can inhibit this heterodimer and exert the opposite function [22,150]. After NEAT1-2 has been transcribed completely, RNase P can recognize the t-RNA-like structure and cleaves it to form the non-polyadenylated 3′-end of NEAT1-2 [157].

In breast cancer, ALYREF can stabilize CPSF to elevate the NEAT1-1 isoform, promoting carcinogenesis [158]. Furthermore, another important protein, PTBP3, can increase the expression of NEAT1-2 and total NEAT1 but decrease the induction of miR-612, which has an overlapping sequence with NEAT1-2, as mentioned above. Finally, PTBP3 promotes the growth and metastasis of hepatocellular carcinoma [23]. Obviously, NEAT1-1 almost always shows an oncogenic function in cancers excluding AML. However, the function of NEAT1-2 in cancer is controversial and there are only a few relevant studies. In colorectal cancer, a study showed that NEAT1-2 can inhibit caner proliferation [159]. In contrast, another study showed that a higher level of NEAT1-2 indicates a poor survival rate and can promote proliferation and glycolysis in hepatocellular carcinoma [160]. NEAT1-2 has also been shown to promote the metastasis of thyroid cancer [35].

Specific proteins are responsible for the stability of NEAT1, including HuR, LIN28B, ALKBH5, PTRF, ALYREF, and SRSF1 [158,161,162,163,164,165]. Specifically, HuR and LIN28B can promote ovarian cancer progression via stabilizing NEAT1 [161,164]. In glioblastoma, PTRF can increase the stability of NEAT1 to promote cancer metastasis [165]. In addition, ALKBH5-stabilized NEAT1 can significantly suppress hypoxia-induced tumor-associated macrophage (TAM) recruitment and immunosuppression in glioblastoma [163]. ALYREF and SRSF1 can also prolong the half-life of NEAT1, facilitating the progression of breast cancer and glioma, respectively [158,162]. In contrast, AUF1 can destabilize NEAT1 and accelerate its decay in HeLa cells [166]. Stability alterations can ultimately affect the expression level of NEAT1 transcripts, thereby regulating cancer progression.

## 9. NEAT1-Containing Exosomes in Cancer

Prokaryotic and eukaryotic cells can generate extracellular vesicles (EVs) and release them into the extracellular environment [167]. EVs can be divided into exosomes and ectosomes. Ectosomes are released via cellular membrane budding with a size range of 50 to 1000 nm, whereas exosomes are generated by endocytosis [168]. The sequential invagination of the cell membrane ultimately forms multivesicular bodies (MVBs), which can intersect with other intracellular vesicles and organelles. When MVBs fuse with the cell membrane, exosomes are released (size range 40 to 160 nm). Exosomes contain many components, including proteins, DNA, RNA, lipids, and metabolites [168].

Exosomal NEAT1 plays an important role in cancers and may serve as a diagnostic or treatment marker. In the exosomes extracted from patients with ovarian cancer, NEAT1 expression levels are upregulated and may contribute to cisplatin resistance [169]. Another study showed that NEAT1 expression is increased in exosomes derived from cancer-associated fibroblasts (CAFs) compared with exosomes derived from normal fibroblasts. Moreover, exosomal NEAT1 derived from cancer-associated fibroblasts (CAFs) promotes the progression of endometrial cancer [79]. Mo et al. found that prostate cancer cell-derived exosomes induce the osteogenesis of mesenchymal stem cells via their NEAT1 component [170]. Additionally, exosomes derived from mesenchymal stem cells pretreated with macrophage migration inhibitory factor (MIF), namely, exosome^MIF^, can alleviate doxorubicin-induced cardiac senescence in chemotherapy [171]. Only a few studies have explored the functions of NEAT1 in exosomes; therefore, the detailed mechanisms of NEAT1 in cancers remain relatively ambiguous.

## 10. Conclusions and Perspectives

NEAT1 generally exerts oncogenic effects, promoting the progression of various cancers, excluding AML, in which NEAT1 facilitates the differentiation of leukocytes to mitigate the progression of the disease. NEAT1 can bind with miRNAs to upregulate the corresponding mRNAs in the cytosol via a ceRNA mechanism. Moreover, NEAT1 can interact with certain transcription factors, enzymes, or nuclear receptors to regulate transcription, histone modification, and protein stability. NEAT1 exerts its guide function by recruiting these proteins and exerts its decoy function by sequestering these factors. NEAT1 can also exert its scaffold function by providing a reactive platform for these molecules. Consequently, many tumorigenic factors are elevated, facilitating the progression of cancers.

There are many studies related to NEAT1, but how NEAT1 plays the role of ceRNA remains ambiguous. NEAT1 is transcribed in the nucleus and is involved in the assembly of paraspeckles. However, NEAT1 only exerts its sponge function after it is transported out of the nucleus. Further studies are needed to determine which proteins containing nuclear export sequences can interact with NEAT1. Consequently, NEAT1 is translocated through the nuclear pore complex under the influence of a specific exportin and enters the cytoplasm. In addition, there is a close relationship between paraspeckles and cell differentiation, but researchers scarcely focus on the formation of paraspeckles in cancers. Chen et al. reported that paraspeckles are absent in undifferentiated embryonic stem cells, but the number of paraspeckles gradually increases after the induction of differentiation [172]. The paraspeckles can retain many molecules related to cell stemness maintenance, thereby inhibiting cell stemness [156]. Therefore, not only the knockdown or knockout of NEAT1 but also the induction of paraspeckles may play an enlightening role in the future treatment of cancer.

## Figures and Tables

**Figure 1 cancers-14-04009-f001:**
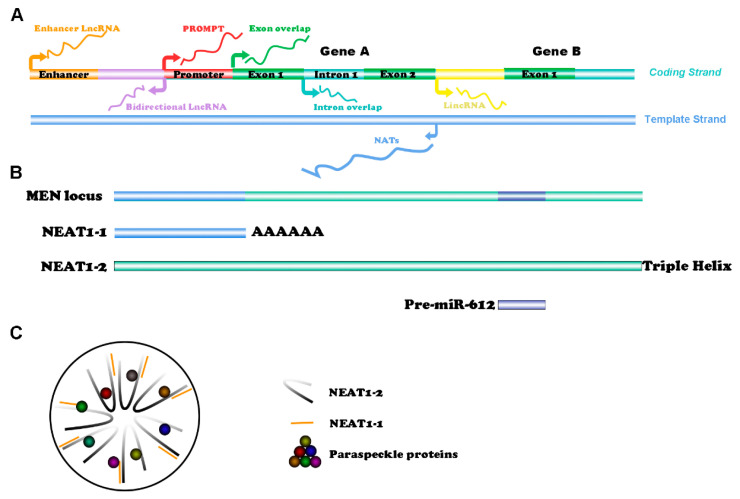
Classification of lncRNAs and structure of NEAT1: (**A**) LncRNAs can be categorized as enhancer lncRNAs, promoter upstream transcripts (PROMPTs), exon or intron sense-overlapping lncRNAs, long intergenic ncRNAs (lincRNAs), bidirectional lncRNAs, and natural antisense transcripts (NATs); (**B**) Locations of NEAT1-1, NEAT1-2, and Pre-miR-612 at the MEN locus; (**C**) Cross-sectional structure of paraspeckles.

**Figure 2 cancers-14-04009-f002:**
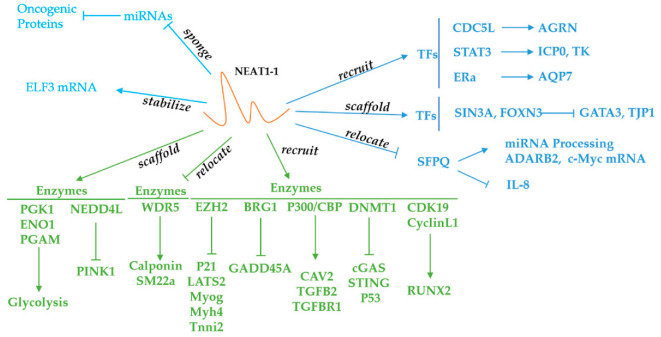
Function and Signaling Pathways of NEAT1. NEAT1 can recruit enzymes and TFs to the genome locus. NEAT1 can also sponge miRs and relocate enzymes and SFPQ to suppress the function of these molecules. In addition, NEAT1 can form a scaffold bridge between different TFs, which allows the enzymes to bind more tightly to the substrates.

**Figure 3 cancers-14-04009-f003:**
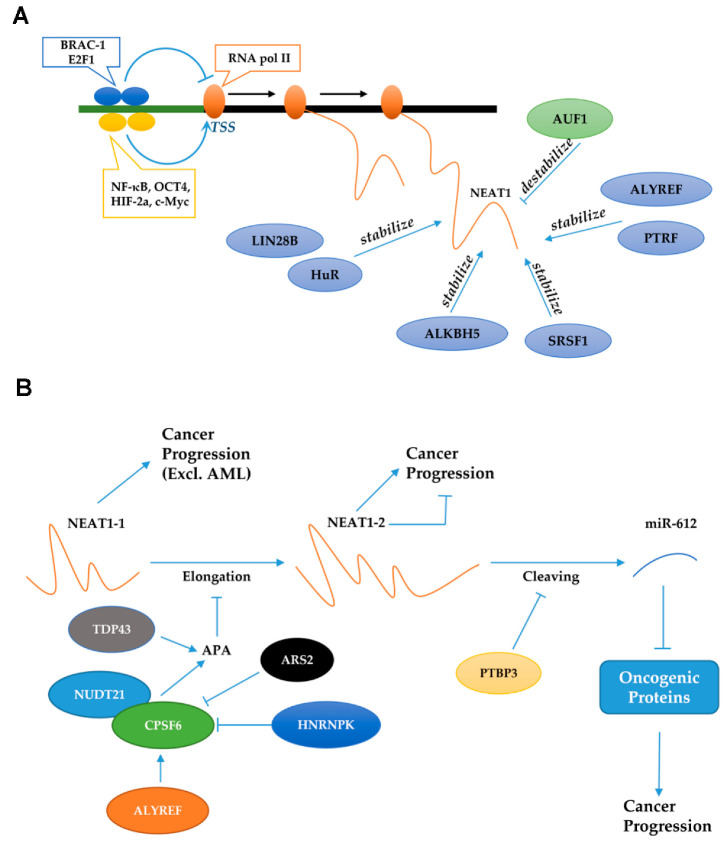
Regulation of NEAT1 at transcriptional level (**A**) and posttranslational level (**B**). (**A**) After being regulated by transcriptional activator or repressor, NEAT1 can also be stabilized or destabilized by some proteins. (**B**) When the polyadenylation (APA) of NEAT1-1 is activated, the transcription site cannot cross the end of NEAT1-1. Consequently, the expression level of NEAT1-1 will be elevated, and the generation of NEAT1-2 will be inhibited. In addition, PYBP3 can inhibit the cleaving of NEAT1-2 to reduce the generation of miR-612, thereby promoting cancer progression.

**Table 1 cancers-14-04009-t001:** NEAT1-miRNA-mRNA network involved in progression of cancer diseases.

Cancer Type	Sponged miRNA	Regulated mRNA	Ref.
Acute Myeloid Leukemia	338-3p	CREBRF	[17]
Breast Cancer	133b	TIMM17A	[94]
	141-3p	KLF12	[41]
	218-5p	TPD52	[95]
	138-5p	ZFX	[40]
	410-3p	Cyclin D1	[38]
	107	CPT1A	[42]
	448	ZEB1	[37]
	211	HMGA2	[39]
	129-5p	WNT4	[96]
Cervical cancer	361	Hsp90	[77]
	124	NF-κB	[97]
	133a	SOX4	[98]
	34a	LDHA	[92]
Cholangiocarcinoma	186-5p	PTP4A1	[78]
Colorectal Cancer	let-7 g-5p	BACH1	[47]
	185-5p	IGF2	[46]
	150-5p	CPSF4	[15]
	196a-5p	GDNF	[44]
	193a-3p	KRAS	[99]
	486-5p	NR4A1β	[100]
	205-5p	VEGFA	[101]
	195-5p	CEP55	[43]
	34a	SIRT1	[102]
Endometrial Cancer	202-3p	TIMD4	[103]
	144-3p	EZH2	[104]
	361	MEF2D, ROCK1, WNT7A, STAT3, VEGFA, PDE4B and KPNA4	[105]
	214-3p	HMGA1	[106]
Esophageal Carcinoma	590-3p	MDM2	[107]
	129	CTBP2	[80]
Gallbladder Cancer	335	Survivin	[81]
Gastric Cancer	365a-3p	ABCC4	[49]
	142-5p	JAG1	[53]
	221-5p	SOX11	[54]
	30a-3p	COX-2, BCL9	[55]
	1224-5p	RSF1	[108]
	1294	AKT1	[48]
	506	STAT3	[52]
	335-5p	ROCK1	[109]
	500a-3p	XBP-1	[51]
Glioblastoma	370-3p	HIF1A	[82]
Glioma	185-5p	DNMT1	[58]
	152-3p	CCT6A	[56]
	139-5p	CDK6	[60]
	132	SOX2	[57]
	449b-5p	c-Met	[59]
Hepatocellular Carcinoma	503	SMO	[63]
	320a	LAGE3	[66]
	22-3p	AKT2	[64]
	296-5p	Calponin 2	[110]
	139-5p	TGF-β1	[65]
	485	STAT3	[61]
	124-3p	ATGL1	[67]
Hodgkin’s Lymphoma	448	DCLK1	[83]
Laryngeal Carcinoma	204-5p	SEMA4B	[84]
Melanoma	495-3p	E2F3	[111]
	200b-3p	SMAD2	[85]
	23a-3p	KLF3	[112]
Multiple Myeloma	214	B7-H3	[113]
Nasopharyngeal Carcinoma	222	ALDH1	[114]
	101-3p	EMP2	[86]
Lung Cancer	204	NUAK1	[70]
	376b-3p	SULF1	[71]
	101-3p	SOX9	[69]
	let-7a	IGF2	[115]
	377-3p	E2F3	[68]
	98-5p	MAPK6	[116]
	26a-5p	ATF2	[117]
	1224	KLF3	[118]
Osteosarcoma	483	STAT3	[87]
	339-5p	TGF-β1	[119]
	186-5p	HIF-1α	[120]
Ovarian Cancer	1321	TJP3	[72]
	let-7 g	MEST	[73]
	4500	BZW1	[74]
	382-3p	ROCK1	[10]
Pancreatic Carcinoma	101	DNA-PKcs	[88]
	302a-3p	RELA	[12]
	335-5p	c-Met	[121]
Prostate Cancer	34a-5p	ACSL4	[76]
	204-5p	ACSL4	[76]
	98-5p	HMGA2	[75]
Retinoblastoma	24-3p	LRG1	[122]
	148b-3p	ROCK1	[123]
	3619-5p	LASP1	[89]
Retinoblastoma	204	CXCR4	[124]
Thyroid Cancer	491	TGM2	[35]
	592	NOVA1	[90]
	129-5p	KLK7	[125]
	106b-5p	ATAD2	[126]
Tongue Carcinoma	339-5p	ITGA3	[91]

**Table 2 cancers-14-04009-t002:** RNA-binding Proteins bound with NEAT1-regulating gene expression.

RBPs	Downstream Target	Function	Ref.
EZH2	H3K27	Trimethylation of H3K27, transcription repression	[131]
WDR5	H3k4	Trimethylation of H3K4, transcription activation	[133]
P300/CBP	H3K27	Acetylation and decrotonylation, transcription activation	[134]
BRG1	H3K27, H3K4	Demethylation of H3K4 and trimethylation of H3K27, transcription inactivation	[135]
DNMT1	*CGAS*, *STING* and *P53* promoter	CpG island methylation, transcription repression	[129]
PGK, PGAM, ENO	1,3-Biphosphoglyceric acid	Glycolysis upregulation	[136]
CDK19, CyclinL1	RNA Pol II	RNA pol II phosphorylation, transcription activation	[137]
NEDD4L	PINK1	Promoting ubiquitination degradation of PINK1	[138]
TRAF6	-	Inhibiting auto-ubiquitination of TRAF6	[139]
STAT3	-	Inhibiting ubiquitination degradation of STAT3	[140]
SIN3A, FOXN3	*GATA3*, *TJP1* promoter	Transcription downregulation	[36]
CDC5L	*AGRN* promoter	Transcription upregulation	[141]
STAT3	*ICP0, TK* promoter	Transcription upregulation	[142]
ERα	*AQP7* promoter	Transcription upregulation	[143]
SFPQ	*IL-8* promoter	Transcription downregulation	[32]
	*ADARB2* promoter	Transcription upregulation	[20]
	Drosha/DGCR8	miRNA processing	[144]
	c-Myc mRNA	Transcription upregulation	[145]
DDX5	β-catenin	Promoting Wnt-β-catenin pathway	[13]
PSPC1	-	IGF1R induction	[146]
SRSF 5	PPAR-γ mRNA	mRNA splicing	[147]
U2AF65	hnRNP A2 mRNA	Increasing hnRNP A2 expression	[148]

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
