# Peer review of "Molecular Interactions of the Long Noncoding RNA NEAT1 in Cancer"

_cancers, 2022, doi:10.3390/cancers14164009_

Round 1

Reviewer 1 Report

All of my points have been answered to my satisfaction.

Author Response

Thanks for your recognition.

Reviewer 2 Report

The authors reviewed the molecular interactions of NEAT1 in cancer. However, the manuscript had several problems that deduct the quality of this manuscript. Specific comments are listed below.

1. The abstract merely mentioned what the aim is, but no implications and/or why this review is important.

2. What is the figure between fig2 and “5. NEAT1-binding enzymes” ?

3. English editing is required. For example, line 364-365” Generally, transcription factors can bind to the promoter region of NEAT1, thereby elevating the expression of NEAT1.” That is what transcription factors do. Not sure why mentioned it here like transcription factors only activate expression of NEAT1.

4. Lack of figure legends. It is important that each figure is self-explanatory.

5. Lack of clinical potential and future research direction of NEAT1.

6. I am confused by the gene and proteins mentioned in this study. Please use italic to differentiate.

7. line 178: “Upregulation of NEAT1 in cancer tissues has been observed in many studies.” What are the studies? No references are provided here. Also, if this is true, why only mentioned this in lung cancer section?

8. A main problem of this review is that it is difficult to tell that it is author’s idea or from the literature. Also, the authors tend to ignore references. Therefore, the authors should work hard on the references.

I list one examples:

“Similarly, BRG-1, which is a subunit of SWI/SNF, can interact with NEAT1; thus, BRG-1 can be recruited to the promoter region of GADD45A, increasing H3K27me3 and decreasing H3K4me3. As a result, the transcription of GADD45A is inactivated, and gastric cancer is aggravated [135].”

=> Are the two sentences all the result from ref#135?  Is “BRG-1, which is a subunit of SWI/SNF” the result of ref#135?  Is “BRG-1 can interact with NEAT1” also a result of ref#135?

Author Response

Response Letter

Thanks for your comments concerning our manuscript. Those comments are all valuable and very helpful for revising and improving our paper. We have studied comments carefully and have made correction which we hope meet with approval. The main corrections in the paper and the responds to the reviewer’s comments are as following:

Reviewer #2

  1. The abstract merely mentioned what the aim is, but no implications and/or why this review is important.

Response: Thanks for your suggestions. Implications of NEAT1 in cancer diseases has been added in the abstract.

“Hence, elucidating the molecular interaction of NEAT1 may shed light on future treatment of cancer.” See change-tracked manuscript, line 13-14.

  1. What is the figure between fig2 and “5. NEAT1-binding enzymes” ?

Response: That figure had been deleted and marked when we submitted the manuscript in the first time. If you accept the revision in Word Office, that figure will disappear.

  1. English editing is required. For example, line 364-365” Generally, transcription factors can bind to the promoter region of NEAT1, thereby elevating the expression of NEAT1.” That is what transcription factors do. Not sure why mentioned it here like transcription factors only activate expression of NEAT1.

Response: Thanks for your meticulous suggestions. We have rephrased our description to reduce ambiguity.

“Studies have shown that some transcription factors can directly bind to the promoter region of NEAT1 to regulate its expression.” See change-tracked manuscript, line 368-369.

  1. Lack of figure legends. It is important that each figure is self-explanatory.

Response: Thanks for your meaningful suggestions. Figure legends of figure 2 and figure 3 have been added.

“Figure 2. Function and Signaling Pathways of NEAT1. NEAT1 can recruit enzymes and TFs to the genome locus. NEAT1 can also sponge miRs, relocate enzymes and SFPQ to suppress the function of these molecules. In addition, NEAT1 can form a scaffold bridge between different TFs, and it allows the enzymes to bind more tightly to the substrates.”

See change-tracked manuscript, line 241-244.

“Figure 3. Regulation of NEAT1 in transcriptional level (A) and posttranslational level (B). (A) After regulated by transcriptional activator or repressor, NEAT1 can also been stabilized or destabilized by some proteins. (B) When the polyadenylation (APA) of NEAT1-1 is activated, the transcription site cannot cross the end of NEAT1-1. Consequently, the expression level of NEAT1-1 will be elevated and the generation of NEAT1-2 will be inhibited. In addition, PYBP3 can inhibit the cleaving of NEAT1-2 to reduce the generation of miR-612, promoting cancer progression.”

See change-tracked manuscript, line 424-429.

  1. Lack of clinical potential and future research direction of NEAT1.

Response: Thanks for your suggestions. Clinical potential and future research direction of NEAT1 have been supplemented

“There are many studies related to NEAT1, but how NEAT1 plays the role of ceRNA remains ambiguous. NEAT1 is transcribed in the nucleus and involved in the assembly of paraspeckles. However, NEAT1 only exerts its sponge function after transported out of the nucleus. Further studies are needed to determine which pro-teins containing nuclear export sequences can interact with NEAT1. Consequently, NEAT1 is translocated through the nuclear pore complex under the influence of spe-cific exportin and enters the cytoplasm. In addition, there is a close relationship be-tween paraspeckles and cell differentiation, but researchers scarcely focus on the for-mation of paraspeckles in cancer diseases. Chen et al. reported that paraspeckles are absent in undifferentiated embryonic stem cells, but the number of paraspeckles gradually increases after the induction of differentiation [172]. The paraspeckles can retain many molecules related to cell stemness maintenance, thereby inhibiting cell stemness [156]. Therefore, not only the knockdown or knockout of NEAT1, but also the induction of paraspeckles may play an enlightening role in the future treatment of cancer.” See change-tracked manuscript line 553-566.

  1. I am confused by the gene and proteins mentioned in this study. Please use italic to differentiate.

Response: Thanks for your suggestions. In Homo sapiens, protein names have been indicated in capital letters and gene names have been indicated in italic capital letters.

For example:

“multiple endocrine neoplasia (MEN) locus” See change-tracked manuscript line 36.

CGAS, STING and P53” See change-tracked manuscript line 254.

P21, LATS2, MYOG, MYH4 and TNNI2” See change-tracked manuscript line 262.

“promoter region of AGRN” See change-tracked manuscript line 300.

  1. line 178: “Upregulation of NEAT1 in cancer tissues has been observed in many studies.” What are the studies? No references are provided here. Also, if this is true, why only mentioned this in lung cancer section?

Response: We have rephrased our description and added the corresponding references.

“Upregulation of NEAT1 in lung cancer tissues has been observed in several studies [67, 68, 70]” See change-tracked manuscript line 175-176.

  1. A main problem of this review is that it is difficult to tell that it is author’s idea or from the literature. Also, the authors tend to ignore references. Therefore, the authors should work hard on the references.

I list one examples:

“Similarly, BRG-1, which is a subunit of SWI/SNF, can interact with NEAT1; thus, BRG-1 can be recruited to the promoter region of GADD45A, increasing H3K27me3 and decreasing H3K4me3. As a result, the transcription of GADD45A is inactivated, and gastric cancer is aggravated [135].”

=> Are the two sentences all the result from ref#135?  Is “BRG-1, which is a subunit of SWI/SNF” the result of ref#135?  Is “BRG-1 can interact with NEAT1” also a result of ref#135?

Response: Thanks for your suggestions. These two sentences are also the result of ref#135. We have added the ignored ref#135.

“Similarly, BRG-1, which is a subunit of SWI/SNF, can interact with NEAT1; thus, BRG-1 can be recruited to the promoter region of GADD45A, increasing H3K27me3 and decreasing H3K4me3 [135].” See change-tracked manuscript line 278-280.

We also added other ignored references.

Ref 44, 45. See change-tracked manuscript, line 139

Ref 49-51. See change-tracked manuscript, line 146

Ref 8. See change-tracked manuscript, line 165

Ref 130. See change-tracked manuscript, line 254

Ref 132. See change-tracked manuscript, line 261

Ref 92. See change-tracked manuscript, line 272

Ref 134. See change-tracked manuscript, line 276

Ref 139. See change-tracked manuscript, line 283

Ref 138. See change-tracked manuscript, line 287

Ref 36. See change-tracked manuscript, line305

Ref 137. See change-tracked manuscript, line 355

Ref 152. See change-tracked manuscript, line361

Ref 168. See change-tracked manuscript, line 523

Round 2

Reviewer 2 Report

The authors tried to address most of my comments.

This manuscript is a resubmission of an earlier submission. The following is a list of the peer review reports and author responses from that submission.

Round 1

Reviewer 1 Report

The authors aim to review the molecular interactions of the noncoding RNA NEAT1 in cancer. A better understanding of NEAT1 is important because it has emerged as an oncogenic factor in many cancer types. However, the manuscript is poorly written and poorly organized. Specific comments are listed below.

  1. Both the abstract and introduction are not structured. The main idea is missing is the abstract. As a reviewer paper, the introduction fails to provide general introduction to NEAT1 and the structure of the manuscript.
  2. References were wrongly cited. For example: The current references (#1 and #2) are specific to NEAT1, not the general statement described here. The authors should double check all the references mentioned in the manuscript.
  3. As a review paper, it should cite the original references, not other review papers when representing a specific finding. The authors should check this issue completely. For example, #19 should not be cited here; instead, the correct one should be cited is Prasanth KV, Prasanth SG, Xuan Z, Hearn S, Freier SM, Bennett CF et al (2005) Regulating gene expression through RNA nuclear retention. Cell 123(2):249–263.
  4. “8. Factors that Regulate NEAT1 Expression” should move to the earlier section. Are these factors impact cancer development and progression? No studies or cancer types were specified here. Also a figure showing the regulation is needed.
  5. Are NEAT1-1 and NEAT1-2 expressed at the same amount? A figure representing the expression of the two isoforms across cancer types should be provided (ex: TCGA data and other studies). miR612 should also be included.
  6. Any differences between NEAT1-1 and NEAT1-2 in terms of their regulation and regulatory patterns/effects?
  7. In my opinion, “9. Function of NEAT1 in Various Cancers” should be broken down and put where the interaction mechanism is. Now this section merely summarizes the association. In the current format/writing, the readers have to go back to the original reference to get the idea. The findings should be paired with the regulatory roles of NEAT1. For example, many of the references mentioned here are already mentioned in previous sections. I would be more interested in which regulatory role of NEAT1 played in which cancer and what’s the effect. For example, which pathways were affected and what’s the consequences?
  8. The paper missed other known RBPs that interact with NEAT1. A more thorough survey is required. For example, FUS/TLS (PMID: 27147820).
  9. Metabolic reprogramming and immune microenvironment are two important features of cancer progression. NEAT1 has been shown to be involved in both of them, but the manuscript fails to include them.
  10. Figure resolution needs to be improved. The current version is difficult to read. Also, Figure 2 should be more informative and provides more details. It’s more like a graphical abstract now, not a figure trying to specify regulatory patterns of NEAT1.
  11. The following sentence is incorrect because A->G change in the coding regions does not necessarily result in protein recording. It could be synonymous substitution. “Generally, dsRNA-dependent adenosine deaminases (ADARs) can convert adenosine (A) into inosine (I), which will be identified as guanosine (G) in the translation process, forming a different peptide [24].”
  12. Professional English editing service is required. The paper is not scientifically structured and not concise. Each main topic lacks general introduction and conclusion.

Reviewer 2 Report

The current manuscript reviews the updated information on the interactions of Neat1 lncRNA with other factors. Such concerted efforts are important in understanding the possible mechanism of action of lncRNA under focus. However, I have the following few observations/suggestions on this current review:

  1. The authors need to highlight very clearly what new information/idea they are proposing or articulating about a well studied and published lncRNA like Neat 1.
  2. Are there any cancer specific Neat1 interactions that are different from those in disease types? This would be justify the title and focus on cancer in particular in this current manuscript rather than focusing on the mechanism for the lncRNA in general.
  3. The figures in the manuscript needs to improve. Reviews should have more informative, illustrative higher quality figures. 
  4. Does this current review is aiming to just add/update to the names of molecular interactions to the 2018 review of the same lncRNA rather than adding any new interpretation/mechanisms of interaction based on publications from 2018-2022? 
  5. A model of proposed article objective or modelling of the main outcome  /summary from this current review would be interesting for reader's understanding. 

Reviewer 3 Report

This is a very well written and thorough review of a very versatile lincRNA.
However, it would benefit from a little more discussion, because I think NEAT1 function is very confusing and it would be nice to have that discussed. 
(As an example, every cancer type appears to have a distinct pattern of miRNA NEAT1 binding, where even the same miRNA then controls a different protein. With one exception, this leads to a negative effect of NEAT1 expression. ) 

I also have some minor points:
Line 22: Please cite ENCODE for the 2% translated into proteins.
Line 34: (37) You write that elevated NEAT1 always indicates poorer survival rate    But in APL this seems to be a protective factor - would almost always be better?
Line 84: I would prefer something like "if the mRNA and the miRNA are perfectly complementary, the resulting heterodimers..." because iirc, bulges (unpaired bases) in the duplex lead to repression of translation.

Table 1: The authors of [131] (miR-362 in endometrial cancer also mention VEGF-A and PDE4B as targets - is there a reason that these are not listed here?
Table1: For Retinoblastoma, miR-148b3p and ROCK1, the citation is incorrect. I could not find this miR in [149]. Probably, the authors meant to cite doi: 10.2147/CMAR.S271326 ?
Line 108: NEAT1 and other proteins - please rephrase, as NEAT1 is not a protein.
Line 211: As, interestingly, a triplex prediction was used, I would like you to write something like: by an algorithm analyzing the triplex binding capacity of NEAT1 to DNA duplexes.
Line 249: normal adjacent normal - please rephrase
Line 271: popular: please rephrase
Line 298: Due to the early symptoms are not obvious -> Because/As the early symptoms....
Line 362: If I understood correctly, all Acute myeolid Leukemias (not only APL) are exceptions?